Sex differences in anaerobic performance in CrossFit® athletes: a comparison of three different all-out tests

http://orcid.org/0000-0003-1798-2189 Ponce-García Tomás 1
García-Romero Jerónimo 1
Carrasco-Fernández Laura 1
http://orcid.org/0000-0001-8524-1847 Castillo-Domínguez Alejandro 2
http://orcid.org/0000-0001-7546-7965 Benítez-Porres Javier 1 3 4 benitez@uma.es
1 Human Physiology, Physical Education and Sports, Universidad de Málaga , Málaga , Spain
2 Department of Nursing and Podiatry, Universidad de Málaga , Málaga , Spain
3 Instituto de Investigación Biomédica de Málaga y Plataforma en Nanomedicina-IBIMA Plataforma BIONAND , Málaga , Spain
4 CIBER Fisiopatología de la Obesidad y Nutrición (CIBEROBN), Universidad Carlos III de Madrid , Madrid , Spain
Penna Eduardo
Electronic publication date: 2025 Feb 5
Publication date: 2025
Volume: 13
Electronic Location ID: e18930
Received 2024 Sep 9; Accepted 2025 Jan 13
Copyright: © 2025 Ponce-García et al.
Copyright year: 2025
Copyright holder: Ponce-García et al.
License: This is an open access article distributed under the terms of the Creative Commons Attribution License, which permits unrestricted use, distribution, reproduction and adaptation in any medium and for any purpose provided that it is properly attributed. For attribution, the original author(s), title, publication source (PeerJ) and either DOI or URL of the article must be cited.
License URL: https://creativecommons.org/licenses/by/4.0/

Keywords: Sports performance, Body composition, Athletes, CrossFit®, High-intensity functional training

Funding: The authors received no funding for this work

==============================
Background

Athletic performance can be influenced by various factors, including those related to biological sex. Various scientific disciplines have studied the observed differences in athletic performance between men and women. Moreover, anaerobic performance refers to the capacity of the human body to generate energy quickly and efficiently during high-intensity and short-duration activities. It is associated with the ability to perform explosive actions and the capacity for rapid recovery between repeated efforts. Anaerobic performance is a determining factor for performance in high-intensity sports and those with predominantly lower intensity but intermittent peaks of higher intensity. One high-intensity sport that has experienced exponential growth and attracts increasing numbers of participants yearly is commercially known as CrossFit® (CF). Therefore, the primary purpose of this study was to determine the anaerobic performance differences between sexes in CF athletes in terms of absolute and relative values.

Methods

A cross-sectional study was conducted over 2 weeks. Fifty CrossFit® athletes (25 men and 25 women) voluntarily participated in the study. They were subjected to body composition analysis and three maximal effort tests to measure anaerobic performance: a cycle ergometer test, a continuous jump test and a squat test.

Results

Significant differences were found in all the variables of absolute peak power and relative to body mass in the three tests. In values adjusted to lean and muscle mass, significant differences were only found in the cycle ergometer test but not in the other two. In mean power variables, significant differences were found in all the variables studied, except for the mean power adjusted to muscle mass in the squat test. In conclusion, this study’s results indicate that differences between sexes in absolute and relative peak powers measured in all tests evaluated are explained by the amount of lean and muscle mass. However, mean powers show significant differences in all variables except for the one related to muscle mass in the squat test.

Introduction

Sports performance may be influenced by various factors, including biological sex. Differences in performance between men and women have been studied in fields such as anatomy, physiology, and sports science (Dominelli & Molgat-Seon, 2022; Hunter, 2016; Lepers, 2019; Lomauro & Aliverti, 2021; Nuzzo, 2023; Rosa-Caldwell & Greene, 2019; Schlegel & Křehký, 2022; Hübner-Wozniak et al., 2004). Increasing interest exists in studying the causes behind these differences, with morphological and physiological characteristics playing a potential role in influencing athletic outcomes. Previous research shows that men generally exhibit higher anthropometric values (weight, height, limb length, muscle perimeters, etc.) than women, which may affect the kinematics and efficiency of sports actions (Podstawski, Markowski & Clark, 2020). Body composition (BC) is another important factor in athletic performance. Men have more muscle or lean mass and less body fat than women on average (Maud & Shultz, 1986; Perez-Gomez et al., 2008). These differences may provide men with an advantage in certain sports, as BC values are related to performance (Mayhew et al., 2001; Potteiger et al., 2010; Stephenson et al., 2015). Additionally, men’s greater absolute strength may be attributed to muscle mass quantity and muscle fiber distribution (Nuzzo, 2022). However, strength (Bishop, Cureton & Collins, 1987) and anaerobic performance (Maud & Shultz, 1986) differences decrease when adjusted for body or lean mass. Furthermore, men have been shown to have a greater high-intensity exertion capacity, possibly due to differences in the utilization of aerobic and anaerobic metabolic pathways (Hill & Smith, 1993), as well as differences in BC (Maciejczyk et al., 2015; Mayhew et al., 2001).

One high-intensity sport that has gained recent popularity for its multidisciplinary approach is CrossFit® (CF). This fitness program focuses on constantly varied functional movements performed at high intensity (Feito et al., 2018), aiming to enhance overall fitness and performance across multiple physical domains. Its comprehensive conditioning method has earned recognition in scientific and sports communities.

High intensity is related to anaerobic performance in athletes due to the physiological adaptations induced by high-intensity training (Franchini et al., 2016). Accurate assessment methods of anaerobic performance are important tools for sports professionals. Specific tests have been developed to measure power and anaerobic capacity. In the laboratory, ergometric tests, such as the Wingate (WG) (Bar-Or, 1987; Smith & Hill, 1991) and in-field tests with vertical jump (Dal Pupo et al., 2014; Nikolaidis et al., 2016) and sprint (Zagatto, Beck & Gobatto, 2009) have been designed to assess power and anaerobic capacity.

Field-based tests have significant advantages over laboratory tests in assessing anaerobic performance in athletes. They simulate more realistic and sport-specific situations and offer greater measurement specificity. These tests are generally more accessible and affordable, allowing more athletes to be evaluated more efficiently. Their lower cost also makes them accessible to coaches with limited budgets or without access to sports medicine laboratories. Due to CF’s multimodal nature and varied movements and exercises, researchers and coaches find it challenging to select a single test to assess performance accurately. Numerous studies have analyzed anaerobic performance using laboratory and field tests across various sports and movements, including cycling, running, and jumping (Andrade et al., 2015; Fry et al., 2014; Miura, 2015; Zagatto, Beck & Gobatto, 2009).

Some studies have examined the relationship between specific laboratory tests and standard CrossFit® workouts (WODs) (Butcher et al., 2015; Dexheimer et al., 2019). Martínez-Gómez et al. (2019) correlated values from a deep squat-based field test to performance in the 2017 CrossFit® Open. Schlegel & Křehký (2022) also compared gender differences in performance during the 2011 CrossFit® Games. The agreement between peak power ratings from various tests in male CF athletes has also been investigated (Ponce-García et al., 2021). However, studies have not been found on sex differences in anaerobic performance across multiple sport-specific tests in these athletes. Therefore, this study aims to identify sex-based performance differences in absolute and relative values of body mass, lean body mass, and muscle mass during maximal effort tests, assess the consistency of these differences across homologous test values, and compare findings with other sports or populations. Understanding these differences may be important for optimizing training program design and scaling.

Materials and Methods

Study design

A cross-sectional study was conducted over 2 weeks. Participants underwent three sessions of performance assessment and one session of BC assessment in the laboratory. In the first session, they underwent BC analysis, and in the subsequent sessions, they underwent maximal exercise testing. All sessions were separated by 48 h.

Participants

Fifty CF athletes participated voluntarily in the present study. Twenty-five males (mean ± SD; age 33.32 ± 5.84 years, height 176.9 ± 4.16 cm, body mass 82.76 ± 7.47 kg) and twenty-five females (mean ± SD; age 33.20 ± 7.78 years, height 162.2 ± 5.01 cm, body mass 62.37 ± 5.50 kg). The sample size was established through statistical power analysis. They were recruited from an advertisement circulated among CF centers owners in Malaga and surrounding areas. Inclusion criteria were that participants had to train at least 3 h per week and had been practicing CF for at least 1 year. All participants were informed of the procedures and provided written informed consent. The procedures in the present study followed the rules of the Helsinki Declaration and were approved by the ethics committee of the University of Malaga.

Anthropometry and body composition

Participants were called to the laboratory for anthropometric measurements and BC analysis in the first session. They were asked to come fasting or not to have eaten or drunk anything for at least 4 h, not to have consumed alcohol in the last 24 h, and not to have taken diuretics in the last week. Their height was measured with a wall-mounted stadiometer accurate to 1 mm (SECA® 206; SECA, Hamburg, Germany), and their mass with a balance accurate to 100 g (SECA® 803; SECA, Hamburg, Germany). Subsequently, BC analysis was performed by dual-energy X-ray absorptiometry (DXA) (Hologic Inc., Bedford, MA, USA) using Hologic APEX software (version 4.6) and electrical bio-impedance (InBody 770, Cerritos, CA, USA). Lean and muscle mass in kilograms were extracted from the BC variables. Muscle mass isolated as a single variable assessed by electrical bio-impedance was included. Its inclusion aimed to compare whether results exclusively on muscle mass, the physiologically active part isolated from lean mass (excluding bones and viscera), made any difference compared to those calculated based on lean mass.

All-out tests

To assess anaerobic performance, three maximal 30-s effort tests were performed: a cycle ergometer test (WG), a repeated jumps test (RJT), and a squat test (AST). The absolute values of peak (PP), mean (XP), and minimum power (MP) were determined for all tests. The fatigue index (FI) was calculated, obtained from the percentage of power loss throughout the test, by the formula: FI (%) = (PP − MP)/PP * 100. Furthermore, relative values were calculated by dividing absolutes values by kilograms of body mass (rPP and rXP), lean mass (rPP.LM and rXP.LM) or muscle mass (rPP.MM and rXP.MM). Participants were asked to refrain from strenuous physical activity 24 h before the all-out tests. Tests were randomly assigned to avoid sequencing bias.

Wingate test

It consisted of 30 s of riding at maximum speed performed using a Monark 894E cycle ergometer (Monark, Vansbro, Sweden) with an applied frictional resistance of 7.5% of body mass for men and 6% for women (Bar-Or, 1987). Participants were asked to ride at 50–70 rpm at 1 kp (50–70 watts) for 10 min to warm up. Subsequently, they took a 5-min recovery interval. Then, at the count of 3, 2, 1… Go! The participant started to ride as fast as possible. A fly-start protocol was used, with a 3-s initial acceleration phase before applying the resistance to begin the test. The researcher encouraged the participant verbally during the test. A 5-min recovery ride at a warm-up pace was set to calm down.

Repeated jumps test

This test consisted of maximum countermovement jumps in 30 s at maximum height. A countermovement jump is a vertical jump initiated in a standing position with both hands on the pelvis by a downward movement. In this jump, the participant rapidly flexes the knees and hips to approximately 90 degrees before explosively extending them to drive the body upward. Participants were previously instructed on the correct way to perform the jumps. For its evaluation, the Chronojump® contact platform (Chronojump Boscosystem, Barcelona, Spain) was used. The absolute power values of each jump were extracted from the Chronojump® software version 2.3. All participants performed 5 min of easy running, three sets of 10 forward jumps, three sets of five vertical jumps, and 5 min of easy running to warm up. Afterwards, a 5-min interval was established to rest and set the platform and software. On the count of 3, 2, 1… Go! The participant began to jump as high and as fast as possible. The researcher verbally encouraged the participant to maintain maximum intensity throughout the interval. After, they walk for 5 min to calm down.

Anaerobic squat test

The test consisted of performing loaded deep squats for 30 s at maximum effort. For the present study, the same test protocol published by Ponce-García et al. (2021) with 75% of the participant’s body mass was used. A standard Olympic lifting set from Xenios USA® (Xenios USA LLC, New York, NY, USA), consisting of a 20 kg bar, plates between 5 kg and 15 kg in increments of 5 kg, and fractional plates of 0.5 and 2.5 kg in increments of 0.5 kg, was used. The recording of barbell speed and power of the repetitions was determined with a Chronojump® linear encoder (Chronojump Boscosystem, Barcelona, Spain). Absolute power values for each repetition were extracted from the data provided by the Chronojump® version 2.3 software. Before each test, participants were weighed to determine the barbell load, rounded to the nearest 0.5 kg. They began the warm-up with 5 min of easy running, followed by ten repetitions with an empty barbell, two more sets of ten repetitions with the assigned percentage and ended with 5 min of easy running. Afterwards, a recovery interval of 5 min was set and used to configure the encoder and software. On the count of 3, 2, 1… “Go!” The participant began to work at maximum effort, trying to perform as many squats as possible and being verbally motivated by the researcher throughout the test. They were asked to walk for 5 min to recover.

Statistical analysis

Table 1 presents the BC descriptive data of the sample in mean values and standard deviations. The normality of the variables was analyzed using the Shapiro-Wilk test, and the homogeneity of variances using the Levene test. A logarithmic transformation was performed for those variables that did not comply with some of the above assumptions (TRFM and WBFM in males). The Student’s t-test was used to compare the mean values of the groups. SPSS version 21.0 software (IBM Corp., Armonk, NY, USA) was used to perform the different analyses. The effect size measure for sex differences was determined using Cohen’s d by calculating the difference in means between the two sexes and dividing the result by the pooled standard deviation. The magnitude of effect sizes was interpreted; d < 0.2 was considered no effect, 0.2 ≤ d < 0.5 was considered a ‘small’ effect size, 0.5 ≤ d < 0.8 represented a ‘medium’ effect size, and d ≥ 0.8 was considered a ‘large’ effect size (Cohen, 1988). The significance level was set at p < 0.05. In addition, differences between males and females were calculated as percentages through the following formula: %-dif = (MV − FV)/MV * 100, where MV corresponds to male values and FV to female values. Positive percentage values mean higher in men and negative percentage values mean higher in women.

Table 1 Descriptive data of the variables, gender comparison and effect size.

	Group (n = 50)	Male (n = 25)	Female (n = 25)			
	Mean	SD	SEM	Mean	SD	SEM	Mean	SD	SEM	t	d	
Age (years)	33.26	6.81	0.96	33.32	5.84	1.17	33.20	7.78	1.56	0.951	0.02	
Body mass (kg)	72.57	12.17	1.72	82.76	7.47	1.49	62.37	5.50	1.10	0.000	3.11	
Height (cm)	169.55	8.71	1.23	176.90	4.16	0.83	162.20	5.01	1.00	0.000	3.20	
BMI (kg·m −2 )	25.06	2.31	0.33	26.43	2.03	0.41	23.70	1.70	0.34	0.000	1.46	
LM (kg)	58.20	12.15	1.72	69.01	6.15	1.23	47.39	4.46	0.89	0.000	4.02	
FM (kg)	15.49	3.14	0.44	15.15	3.25	0.65	15.82	3.05	0.61	0.459	0.21	
FM (%)	21.45	4.95	0.70	17.95	3.09	0.62	24.96	3.86	0.77	0.000	2.01	
SMM (kg)	34.32	7.59	1.07	41.08	3.73	0.75	27.55	2.88	0.58	0.000	4.06	
Note:

SD, standard deviation; SEM, standard error of the mean; t, significance of t-student analysis; d, Cohen d effect size; BMI, body mass index; LM, whole body lean mass in kg; FM, whole body fat mass in kg; FM, whole body fat mass as percentage; SMM, skeletal muscle mass in kg; %-dif, percentage of difference.

Results

Table S1 shows absolute power values, adjusted for body mass, lean mass, muscle mass and fatigue index from each test. T-test results, effect sizes (d) and percentages of differences between sexes are also shown. Differences are graphically represented in Fig. 1.

Figure 1 Comparison between sexes of peak and mean absolute and relative powers.

WG, Wingate test; RJT, repeated jump test; AST, anaerobic squat test; (A) absolute peak power in watts (W); (B) peak power relative to body mass in watts per kg (W/kg); (C) peak power relative to lean mass in watts per kg (W/kgLM); (D) peak power relative to muscle mass in watts per kg (W/kgMM); (E) absolute mean power in watts (W); (F) mean power relative to body mass in watts per kg (W/kg); (G) mean power relative to lean mass in watts per kg (W/kgLM); (H) mean power relative to muscle mass in watts per kg (W/kgMM); *significant differences at p < 0.05 level.

Peak power

Significant differences were observed in all absolute variables (PP) and those adjusted for body mass (rPP) across all three tests (p < 0.01). When adjusting for lean mass (rPP.LM), significant differences were noted in the WG test (p < 0.01), whereas the RJT (p = 0.186) and AST (p = 0.059) tests did not show significant differences. In muscle mass-adjusted values (rPP.MM), WG also demonstrated significant differences (p < 0.01), while RJT (p = 0.565) and AST (p = 0.206) did not.

Regarding percentage differences, the absolute WG values showed a difference of 43.4% for PP. This percentage decreased to 23.6%, 17.5%, and 15.6% when adjusted for body, lean, and muscle mass, respectively. Notably, all percentage differences were higher in males compared to females.

For the RJT, the absolute percentage difference was 34.6%. When adjusted for body, lean, and muscle mass, the differences were 12.7%, 4.4%, and 1.8%, respectively, with higher values observed in males.

In the AST test, the absolute percentage difference was 36.9%. After adjusting for body mass, this difference was reduced to 16.1%. Further adjustments for lean and muscle mass decreased the differences to 7.6% and 5.6%, respectively, with all percentage differences again being higher in males.

Mean power

Significant differences were observed in all absolute variables (XP), body mass-adjusted variables (rXP), and those related to lean mass (rXP.LM), including WG, RJT and AST (p < 0.01). Specifically, WG, RJT, and AST showed significant differences concerning lean mass (WG: p < 0.01; RJT: p < 0.01; AST: p = 0.015). However, when adjusted for muscle mass (rXP.MM), only WG (p < 0.01) and RJT (p < 0.01) exhibited significant differences, while no significant differences were found for AST (p = 0.057).

The percentage differences in absolute values for WG were 43.3%. After adjusting for body, lean, and muscle mass, these differences decreased to 23.6%, 17.5%, and 15.4%, respectively. For the RJT, the absolute percentage differences were 44.6%, which reduced to 25.9%, 19.2%, and 17.3% after adjusting for body, lean, and muscle mass, respectively. In the AST, the absolute percentage difference was 37.8%, which decreased to 17.5%, 9.3%, and 7.3% when adjusted for body, lean, and muscle mass. Notably, all percentage values were higher in men.

Fatigue index

The only test that demonstrated significant differences between men and women in the FI was the RJT (p < 0.01). No significant differences were observed for the WG (p = 0.235) and the AST (p = 0.067). In percentage terms, men experienced 6.2% more fatigue in the WG than women. Conversely, women exhibited more significant fatigue than men in the RJT (17.5%) and AST (17.2%).

Discussion

The main purpose of the present study was to determine whether there are significant differences between sexes in anaerobic performance in CF athletes in different maximal effort tests.

Peak power

Regarding the absolute peak powers (PP), the present study’s results showed that male athletes exhibited higher values than females, and statistically significant differences were found in all the tests performed (WG, RJT, and AST). The findings are in line with those found by many authors in different populations like untrained university students (Mayhew & Salm, 1990), recreative active adults (Weber, Chia & Inbar, 2006), team sport athletes (Soydan et al., 2018), wrestlers (Hübner-Wozniak et al., 2004), alpine ski racers (Miura, 2015), swimmers (Zera et al., 2022) and sprint cyclists (Ferguson et al., 2023). Moreover, measured in various tests of different nature like cycling (Ferguson et al., 2023; Zera et al., 2022), repeated sprint test (Miura, 2015; Soydan et al., 2018; Hübner-Wozniak et al., 2004), vertical jump (Mayhew & Salm, 1990) and other sports gestures (Mayhew & Salm, 1990; Zera et al., 2022). Although the results have been widely reported by other studies, using absolute values to compare performance between sexes is not the most appropriate due to the morphological and physiological differences and many other factors between men and women that should be considered. Therefore, the absolute values should be normalized or scaled for an optimal comparison between sexes.

In peak power relative to body mass (rPP), like absolute values, men showed higher relative values than women and significant differences in all tests performed (WG, RJT and AST). These differences have also been reported (Vardar et al., 2007; Hübner-Wozniak et al., 2004). Our study’s reduction in differences between sexes agrees with other authors who showed similar results in wrestlers (Vardar et al., 2007; Hübner-Wozniak et al., 2004). These data suggest that adjusting absolute power relative to body mass might be an optimal way to standardize these values. However, it does not consider body composition since different lean or fat mass could show notable differences in relative values in subjects with the same body mass. Thus, significant differences in relative values could suggest that these differences may be related to quality rather than quantity of body mass (Maciejczyk et al., 2015).

When values were adjusted to lean mass (rPP.LM), men also showed higher values than women. However, significant differences were only found in WG but not in RJT or AST. Other authors previously reported this further reduction in differences (Hübner-Wozniak et al., 2004). These results might suggest that peak power is not determined by sex but by other variables, such as body composition (Maciejczyk et al., 2015) or specific tasks.

Relative powers to muscle mass (rPP.MM) likewise showed higher values in men. However, similar to adjusted powers to lean mass, significant differences were only found in WG, not RJT or AST. An additional reduction in the differences between sexes suggests that muscle mass and not lean mass determine peak power. This might be because lean mass includes physiologically non-active tissues such as bone and viscera. Additionally, a significant difference in WG might reveal that men and women use energy substrates or metabolic pathways differently when cycling or due to fibre type distribution or muscle activation differences (Driss & Vandewalle, 2013). The reduction of differences to negligible values, especially in jump and squat tests, again shows that anaerobic power depends on specific exercises and the amount of muscle mass rather than sex.

In percentage terms, the differences in absolute values (PP) between sexes were 43.4% in WG, 34.6% in RJT and 36.9% in AST. However, adjusting values for body mass (rPP) reduced the differences to 23.6% in WG, 12.7% in RJT and 16.1% in AST. This reduction in disparities has been reported previously in other athletes (Hübner-Wozniak et al., 2004). Adjusted values to lean mass (rPP.LM) further reduced differences to 17.5% for WG, 4.4% for RJT and 7.6% for AST. Lastly, relating values to muscle mass (rPP.MM) further reduced the differences to 15.6%, 1.8% and 5.6% in WG, RJT and AST, respectively. The reductions in differences found in values adjusted for body, lean and muscle mass suggest that peak power is determined mainly by muscle mass quantity. However, those differences differ depending on the assessed sporting gesture.

These results show that men have higher absolute and relative peak power values in all tests, regardless of BC. The decrease in differences with the adjustment of the values to lean mass or muscle mass indicates that sex is not as determining as these variables in anaerobic power in CF athletes. Likewise, these differences vary in the different assessed tasks.

Mean power

The absolute mean power values (XP) were higher in male athletes. They showed significant differences between sexes in all tests, as previously reported by Soydan et al. (2018), Hübner-Wozniak et al. (2004), and Zera et al. (2022). Like in peak powers, other physiological or morphological factors may determine differences in absolute values.

In mean powers related to body mass (rXP), values were higher in men, and their differences remained significant in all tests. However, the adjustment reduced differences between the sexes. This reduction has been previously published by other authors (Hübner-Wozniak et al., 2004). However, adjusting absolute power values to body mass may not be appropriate since it does not consider other variables, such as body composition, that could be directly related to the ability to produce greater power or sustain maximum effort for longer.

Values relative to lean mass (rXP.LM) continued to be higher in men, and significant differences in all tests were still present, similar to those reported in previous studies (Hübner-Wozniak et al., 2004). Furthermore, after eliminating body fat, persistent differences between groups may suggest that physiological issues such as better use of glycolytic metabolic pathways by male athletes may determine differences between sexes (Esbjörnsson et al., 1993).

Adjusted values to muscle mass (rXP.MM) were still higher in males but showed significant differences in WG and RJT, not in AST. These results may indicate that the ability to sustain maximal effort could be directly related to the contractile or metabolic properties of skeletal muscle, the higher proportion of type II fibres, and the greater capacity to regenerate ATP anaerobically in men (Esbjörnsson et al., 1993). Likewise, kinematic differences between sexes may determine the differences in power values adjusted to muscle mass in RJT (Kernozek, Torry & Iwasaki, 2008; Pappas et al., 2007).

In percentages, the difference between male and female athletes in absolute values (XP) was 43.3% in WG, 44.6% in RJT and 37.8% in AST. When values were related to body mass (rXP), those differences were reduced to 23.6%, 25.9%, and 17.5% in WG, RJT and AST, respectively. Subsequently, a more significant reduction occurred when values were adjusted for lean mass (rXP.LM): 17.5% in WG, 19.2% in RJT, and 9.3% in AST. Finally, relating values to muscle mass (rXP.MM) further reduced the differences to 15.4% in WG, 17.3% in RJT, and 7.3% in AST.

These results show that men exhibit higher mean absolute and relative power values in all tests. The differences observed in all tests, except the muscle mass-adjusted values in AST, suggest that the ability to sustain a maximum effort differs between sexes, possibly due to metabolic factors in WG or kinematics in RJT.

Fatigue index

Concerning FI, our results showed statistically significant differences in RJT but not in WG or AST. The differences found in RJT could be attributable to biomechanical or neuromuscular factors (Kernozek, Torry & Iwasaki, 2008; Márquez et al., 2017; McMahon, Rej & Comfort, 2017; Pappas et al., 2007). In contrast, no such differences were found in WG or AST. Similar fatigue indices between sexes have been previously reported by other authors in team athletes in a repeated sprint test (Soydan et al., 2018).

Another notable finding is that men showed 6.2% more fatigue in the WG than women. However, females exhibited more fatigue in the RJT and AST than males, 17.5% and 17.2%, respectively. These data suggest that fatigue might depend on the task’s specific demands (Kernozek, Torry & Iwasaki, 2008; Pappas et al., 2007; Pappas & Carpes, 2012; Hübner-Wozniak et al., 2004).

We would like to highlight some of the present study’s strengths. First, the sample size is considerably larger than the average of studies in this field. Second, the study compares three different tests based on different sports actions. This makes the applicability of the results of this study more comprehensive, especially in field-based tests, which offer sports professionals tools for immediate application to CF athletes.

Some limitations can be recognized in the present study. For example, gas exchange records for all tests are not available to determine whether differences in fatigability might be related to anaerobic metabolism, kinematic data that might be related to differences in performance in the different anaerobic tests, or data related to the participants’ menstrual cycles that might affect the performance of the female athletes.

Conclusions

In conclusion, the results of this study indicate that the differences between sexes in absolute and relative peak and mean powers, as well as the amount of lean mass and muscle mass, explain power outputs assessed in both laboratory and field tests. The reduction of these differences in the values relative to lean and muscle mass suggests that anaerobic performance depends more on the amount of muscle mass. From this, training programs designed to increase lean muscle mass might benefit, to some extent, anaerobic performance in CF athletes, particularly in female athletes. However, further research is needed to understand better body mass and optimal lean and muscle mass levels that enhance anaerobic performance without detriment to the athletes’ other abilities.

Supplemental Information

Supplemental Information 1 Descriptive data of all absolute and relative performance values, gender comparison, effect sizes and percentage of differences.

Supplemental Information 2 STROBE Statement.

Supplemental Information 3 Raw data.

The authors wish to thank Teatinos Functional Fitness and all participants for their collaboration and the support from the University of Málaga (Campus of International Excellence Andalucía Tech) in providing facilities.

Additional Information and Declarations

Competing Interests

The authors declare that they have no competing interests.

Author Contributions

Tomás Ponce-García conceived and designed the experiments, performed the experiments, analyzed the data, prepared figures and/or tables, authored or reviewed drafts of the article, and approved the final draft.

Jerónimo García-Romero conceived and designed the experiments, authored or reviewed drafts of the article, and approved the final draft.

Laura Carrasco-Fernández performed the experiments, authored or reviewed drafts of the article, and approved the final draft.

Alejandro Castillo-Domínguez performed the experiments, authored or reviewed drafts of the article, and approved the final draft.

Javier Benítez-Porres conceived and designed the experiments, analyzed the data, prepared figures and/or tables, authored or reviewed drafts of the article, and approved the final draft.

Human Ethics

The following information was supplied relating to ethical approvals (i.e., approving body and any reference numbers):

Comité Ético de Experimentación de la Universidad de Málaga (CEUMA: 43-2018-H)

Data Availability

The following information was supplied regarding data availability:

The raw data is available at the University of Málaga: https://dx.doi.org/10.24310/riuma.35168.

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
