# Peer review of "Sex differences in anaerobic performance in CrossFit® athletes: a comparison of three different all-out tests"

_PeerJ, doi:10.7717/peerj.18930_

## Round 0.1 · original submission · Major Revisions

I concur with the reviewers that your manuscript requires further revision to be suitable for publication in PeerJ. Although both reviewers acknowledge that the methodology is sufficiently robust to address the research question, all sections (introduction, methods, results, and particularly the discussion) should be revised to enhance clarity and improve overall understanding.

I would also like to emphasize that both reviewers have expressed concerns about the acceptance of this manuscript in its current form. However, after addressing all the major revisions requested by the reviewers, the manuscript has the chance to meet the journal's quality standards.

Reviewer 1 ·

Basic reporting

The manuscript entitled " Sex differences in anaerobic performance in CrossFit athletes: A comparison of three different all-out tests" to determine the existing performance differences between sexes in absolute and relative values for body mass, lean body mass, and muscle mass in different tests at maximal effort, whether the differences differ between homologous values in various tests and compare the similarity of these possible differences with those found in other sports or populations. The idea is original, and the sample studied (50 volunteers) is considerable. However, some aspects do not allow me to approve this manuscript version. Several concerns are raised about the introduction, results, and discussion need to be improved in a future version of this manuscript.

Experimental design

The experimental design of the present study is suitable for answering the research question.

Validity of the findings

Although the findings of this study have broad application in the sports context and may benefit assessment procedures as well as the practice of recreational and competitive sports practitioners with anaerobic demands, the authors fail to highlight this point in the text. Furthermore, the discussion is not very expanded and does not provide a generalization of the results or applicability of the findings.

Additional comments

Introduction
General comment: The introduction section is too extensive. The authors should be more concise. Connective elements are overused and the link between complementary information could be improved. The authors do not provide sufficient literary support to justify the analysis of power outputs in relation to lean and fat muscle mass. Thus, the authors should include the reason for this division.

Line 50: "Observed differences in athletic performance between men and women have been the subject of study in several scientific disciplines". Authors must cite which disciplines.

Line 54: Long-winded sentence. The authors could be more concise, for example: "Morphological and physiological characteristics vary between the sexes and may influence athletic performance" instead of "These differences can be attributed to various morphological and physiological characteristics that vary between the sexes and may influence athletic performance."

Line 64: Another example of long-winded sentence. "In addition to higher levels of muscle mass, the greater manifestation of absolute muscle strength exhibited by men compared to women may be due to differences in the distribution of muscle fibers (Nuzzo, 2022)". Authors could be more direct in their writing to make reading more fluid. Too many connective and appositive elements make comprehension worse.

Line 78: This sentence doesn't sound complete. Rewrite it:
"High intensity is also closely related to anaerobic performance in athletes."

Line 85: To improve reading fluency, these two sentences should be inverted:

"Field tests have significant advantages over laboratory tests in assessing anaerobic performance in athletes. By simulating more realistic and sport-specific situations, they offer greater measurement specificity."

Line 90: The sentence "In CF, due to its high-intensity component, anaerobic performance plays an important role." is unnecessary. Also, adjustments should be made in the next sentence to include information about CF.

Line 107: The second part of the aim of this study is confusing. "The authors should rewrite "(...)whether the differences differ between homologous values in different tests and compare the similarity of these possible differences with those found in other sports or populations."

Line 110: I recommend that the authors change the term 'crucial' to 'important'

Methods: Methods were well described and are suitable for answering the research questions. But in several points, the authors could make the text more concise.

Line 142: The sentence is too long. Suggestion: To assess anaerobic performance, three maximal 30-second effort tests were performed, a cycle ergometer test (WG), a repeated jumps test (RJT), and a squat test (AST).

The authors need to give a more detailed description of the three tests used. For example, in Wingate tests the stationary or fly start were used? What is the crank height? In repeated jump tests the mention of “correct way” is not sufficient to describe and allow reproduction of the test if it is of interest to the reader.

Line 173: The authors should not mention the table. It is preferable that cited Descriptive data are mean and SD.

Line 175: What are the variables in which the authors performed a logarithmic transformation? This information must appear in the text.

Results: The authors should consider rewriting some sentences to simplify comprehension and avoid misunderstandings.

Discussion: The discussion is written in a very fragmented manner. This hinders a broad understanding of the research findings. I encourage authors to rewrite the entire discussion of the manuscript.

My suggestion would be to group the eight topics into just three topics or paragraphs. The first would address peak power, the second would address mean power, and the third would address the fatigue index. I believe that an additional paragraph could also be included justifying the difference in the percentage reduction of the different variables analyzed in the study when comparing sexes and tests, to enrich the understanding of the findings.

I also suggest that the authors include the strengths of the study in a section before the limitations of the study. It could be said there that the sample size is considerable, larger than most studies in the area. It could also be said that the tests used have broad applicability in practical contexts and have immediate application for the population in question.

·

Basic reporting

The article presents sufficient introduction and background to demonstrate how the work fits into the broader field of knowledge. However, I suggest some adjustments to be more objective. The subject of the paper was the anaerobic measures, in this sense, it would be important to reduce the text on anthropometric measurements and focus more on the part on anaerobic performance measurements.
The structure of the article had standard sections, however, the numbers in the results and discussion sections should be removed. I also suggest organizing the discussion with fewer subheadings (for example, use only the subheadings “peak power”, “mean power” and “fatigue index”).
Figure 1, table 1 and raw data are appropriate. My native language isn't English, but it seems to me that it still needs improvement to ensure that an international audience can clearly understand your text.

Experimental design

The manuscript presents original research within aims and scope of the journal.
The research question is well defined and is relevant and meaningful. The study contributes to demonstrating the differences in anaerobic performance between men and women with the use of specific tests for crossfit. However, I believe that the great differencial in the study as the use of 3 tests with different motor patterns: 1 test on a cycle ergometer (cycling); 1 test with a jumping squat pattern (power); 1 test with a squat against resistance (75% of body weight). I didn't see, either in the introduction or in the discussion, an approach to the differences in the tests. I believe that this is essential to deepen the discussion.
Methods should be described with sufficient information to be reproducible by another investigator. Rather than citing other authors (as made in the 3 tests), it is important to describe the tests used in the manuscript. Please, bring more information about the tests used in the manuscript.
The investigation had conducted rigorously, to a high technical standard and in conformity with the prevailing ethical standards in the field.

Validity of the findings

All underlying data have been provided; they are robust, statistically sound and controlled. My main concern relates to the discussion and conclusion of the study. The discussion is very descriptive, in other words, it seems like a replication of the results. I'd like to see a discussion focused on the importance of understanding the differences between the sexes in anaerobic performance, especially the question of training guidelines. Some justifications could also be explored for the differences in performance in different standard crossfit wods. Finally, as mentioned above, it would be interesting to discuss the differences in the motor pattern of the 3 tests. For example, the question of whether the difference between men and women remains in the AST even when corrected for muscle mass. Perhaps tests that require maximum strength recruitment may be disadvantageous for women due to the different hormonal environment.
In the conclusion, the importance of increasing muscle mass for better results is extrapolated. I don't know if this conclusion is the most appropriate for this work, not least because it is known that this may not be beneficial.

Additional comments

No comment

---

## Round 0.2 · accepted · Accept

Dear Authors,

First, I would like to sincerely apologize for the delay in responding to the review process for your manuscript. Unfortunately, health issues prevented me from addressing it in a timely manner. I hope this has not caused any inconvenience or negatively impacted your experience with the journal.

I would also like to congratulate you on the quality of your manuscript. I believe all the questions and comments raised by both reviewers were adequately addressed, meeting the journal's standards for publication. This work represents a valuable contribution to the field of CrossFit, which continues to lack robust scientific evidence to inform training prescription and evaluation.

Thank you for your patience and understanding.

·

Basic reporting

The study contributes to demonstrating the differences in anaerobic performance between men and women with the use of specific tests for crossfit. The Introduction section was improved according to the recommendations with the anthropometric measurements text reduced.
The structure of the article had standard sections.

Experimental design

Methods section had a significant improvement with the inclusion of the test’s descriptions. The investigation had conducted rigorously, to a high technical standard and in conformity with the prevailing ethical standards in the field.

Validity of the findings

The Discussion section also had a significant improvement with fewer subheadings and more data discussion. I'd like to see more discussion about the difference in the motor pattern of the tests and the hormonal differences between men and women that could help explain the results found. Do you think they can be added at the end of the discussion?

Additional comments

Abstract
Please, reduce the background section and improve with more information the method section (information about the tests and the measures performed). Add a conclusion section and separate the results section from the conclusion.

Introduction

Lines 66-67, page 7
Can you present more recent papers for the sentence? The papers provided are from 1986 and 1987.

Lines 68-68, page 7
The greater high-intensity exertion capacity of men can be also explained by hormonal differences?

Lines 94-97, page 8
More recent studies have examined the relationship between field tests (related to crossfit) and crossfit workouts. Please, see the papers below

Tibana, R. A., F. H. Dominski, A. Andrade, N. M. F. de Sousa, F. A. Voltarelli, and I. V. de S. Neto. “Exploring the Relationship Between Total Athleticism Score and CrossFit® Open Performance in Amateur Athletes: Single Measure Involving Body Fat Percentage, Aerobic Capacity, Muscle Power and Local Muscle Endurance”. European Journal of Translational Myology, vol. 34, no. 3, Aug. 2024, doi:10.4081/ejtm.2024.12309.

Tibana, Ramires Alsamir et al. “Local Muscle Endurance and Strength Had Strong Relationship with CrossFit® Open 2020 in Amateur Athletes.” Sports (Basel, Switzerland) vol. 9,7 98. 6 Jul. 2021, doi:10.3390/sports9070098

I recommend to insert the references in your text.


Methods

Line 120, page 9
No exclusion criteria?

Lines 131-138, page 9
Why you performed both DXA and bio-impedance? DXA wasn't enough?

Line 193, page 10
I suggest moving table 1 text information to the biggening of the Results section.

Line 197, page 10
Unpaired student t-test? Please add “Unpaired” to the name of the test.


Results

Lines 235-236, page 11
Please, use the correct names (or only the initials) for the tests

Lines 251-253, page 12
The way that is described, it seems that there were differences in FI for WG and AST tests. And it is not true. Please, adjust the text.


Discussion
Lines 256-257, page 12
Why do you not use this objective to the Introduction section? It is so clear in that way. On the other hand, I suggest removing the objective in this site and add a paragraph with the main results before the subtopics of the discussion.

Line 324, page 14
“Absolute mean power values”, not “absolute power values”.